# PROMPT ENGINEERING AND CALIBRATION FOR ZERO-SHOT COMMONSENSE REASONING

**Chenkai Ma**
School of Computer Science and Engineering
University of Electronic Science and Technology of China
Chengdu, 611731, China
kasmas316@gmail.com

## ABSTRACT

Prompt engineering and calibration make large language models excel at reasoning tasks, including multiple choice commonsense reasoning. From a practical perspective, we investigate and evaluate these strategies on smaller language models. Through experiments on five commonsense reasoning benchmarks, we find calibration favors GPT-2 and T5, prompt engineering favors Flan-T5, but their joint effects are mostly negative. [1]

## 1 INTRODUCTION

Large Language models (LLMs) have shown impressive performance in many NLP applications (Ouyang et al., 2022; Chung et al., 2022; Wei et al., 2022a), including commonsense reasoning, a key component to AGI (Davis & Marcus, 2015). Recent studies suggest that LLMs are capable of zero-shot and few-shot learning (Brown et al., 2020; Webson & Pavlick, 2022; Chowdhery et al., 2022), and prompt engineering and calibration can further improve their performance (Kojima et al., 2022; Zhao et al., 2021; Jiang et al., 2021; Kadavath et al., 2022). Despite achieving SOTA performance on many benchmarks, most LLMs are very expensive to use and not released to the public.

Consequently, we study whether prompt engineering and calibration can help smaller language models (those with no more than 3B parameters) in zero-shot multiple choice commonsense reasoning. Since these strategies are likely emergent (Wei et al., 2022b; Chan et al., 2022), we make several modifications, then evaluate them on five commonsense reasoning benchmarks. We find that prompt engineering favors large Flan-T5 models, while calibration works well on GPT-2 and T5. Their joint effects are, however, negative in most cases.

## 2 METHODS

**Background.** Multiple choice commonsense reasoning is formalized as follows: Given a question $x$ and several options $y_1, ..., y_n$, select the best option. In the zero-shot setting, a language model computes a score for each option, which is usually the conditional probability $P_{LM}(y_i|x)$, and selects the one with the highest score, as shown in Figure 1. Recent works suggest that alternatives to the conditional probability can lead to better performance (Holtzman et al., 2021; Niu et al., 2021; Min et al., 2022), but we do not consider these variants for simplicity and fair comparison.

**Prompt engineering: multiple choice prompt and instruction.** A limit of $P_{LM}(y_i|x)$ is that options are not considered jointly. Recent works suggest that providing all the options in the input, along with instructions about the task, is beneficial (Robinson & Wingate, 2023; Chung et al., 2022). Inspired by these ideas, we design templates $T()$ that add an instruction and options to a question, as shown in Figure 1. We do not bind options to symbols like (A), because symbol binding is an emergent ability (Robinson & Wingate, 2023).[2]

---

[1] Code: https://github.com/KasMasVan/Prompt-engineering-and-calibration.
[2] We discuss the effect of symbol binding in Appendix C.

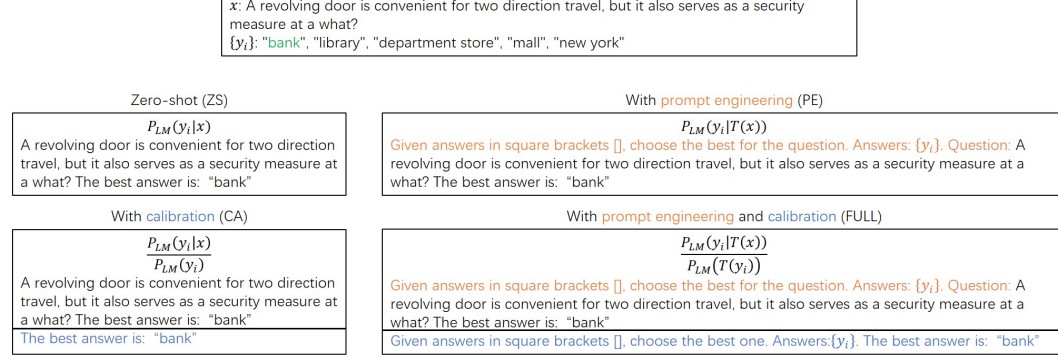

Figure 1: Combinations of data format and option scores for multiple choice commonsense reasoning. Based on the zero-shot method, we add prompt engineering (instruction and multiple choice prompt) and calibration. Unlike previous works, we do not bind options to symbols, like (A).

**Calibration.** Recent works find that language models prefer certain options even without a question, which suggests they are not well-calibrated (Zhao et al., 2021; Jiang et al., 2021). To overcome this problem, we divide the conditional score of an option by another score computed from a "null" prompt that contains no question, as in $\frac{P_{LM}(y_i|x)}{P_{LM}(y_i)}$. An example is shown in Figure 1.

## 3 EXPERIMENTS

**Setup.** We evaluate prompt engineering and calibration on five multiple choice commonsense benchmarks: (1) CommonsenseQA (CSQA) (Talmor et al., 2019); (2) COPA (Gordon et al., 2012); (3) OpenBookQA (OBQA) (Mihaylov et al., 2018); (4)PIQA (Bisk et al., 2019); (5)Social IQA (SIQA) (Sap et al., 2019); We present their statistics in Appendix B. For all benchmarks, we only use their development sets. We compare four zero-shot methods mentioned in Figure 1: (1) ZS, the standard zero-shot method that computes conditional probability scores of each option; (2) CA, which is ZS with calibration, also known as PMI$_{DC}$ in Holtzman et al. (2021); (3) PE, which is ZS with prompt engineering; (4) FULL, which is ZS with both prompt engineering and calibration. As for language models, we use GPT-2 (Radford et al., 2019), T5 (Raffel et al., 2022), and Flan-T5 (Chung et al., 2022). The evaluation metric is accuracy.

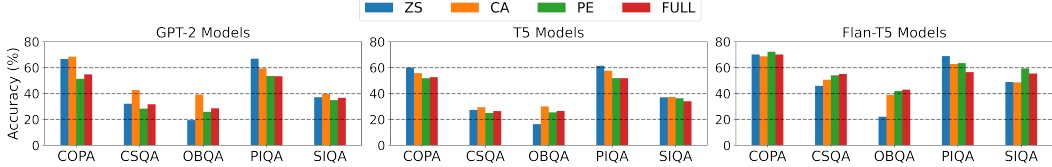

Figure 2: Experiment results on 5 benchmarks, grouped by model families, i.e., GPT-2, T5, Flan-T5.

**Results.** According to Figure 2, We find calibration works best on GPT-2 and T5, and prompt engineering is most performant on Flan-T5. We attribute the former to the surface form competition (Holtzman et al., 2021), and the latter to instruction tuning (Chung et al., 2022). In addition, we find neither strategy works on PIQA, and their joint effects are mostly negative. We leave detailed results and analysis in Appendix C.

## 4 CONCLUSION

We study whether prompt engineering and calibration help smaller language models in multiple choice commonsense reasoning, as they help LLMs. We find that calibration works well on GPT-2 and T5, prompt engineering favors Flan-T5, but their joint effects are mostly negative.

## URM Statement

Author Chenkai Ma meets the URM criteria of ICLR 2023 Tiny Papers Track.

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

## A  FULL PROMPTS FOR ALL BENCHMARKS

In this section, we present prompts (i.e., templates) for each benchmark in Table 1. Specifically, we use one prompt for CSQA and SIQA, and another for COPA, OBQA, and PIQA, because the latter three do not always have a question in a data sample. For simplicity, we still use the term "question" for these three datasets. We also provide the prompts we use for calibration, which is used in FULL.

Table 1: Prompts for each benchmark

| Benchmarks | Prompt for the Question | Prompt for Calibration |
|---|---|---|
| CSQA, SIQA | Given answers in square brackets [], choose the best for the question. Answers: [*answers*]. Question: [*question*] The best answer is: | Given answers in square brackets [], choose the best one. Answers: [*answers*]. The best answer is: |
| COPA, OBQA, PIQA | Given answers in square brackets [], choose the one that best completes the sentence. Answers: [*answers*]. Sentence: [*question*] The best answer is: | Given answers in square brackets [], choose the best one. Answers: [*answers*]. The best answer is: |

## B  DATASET STATISTICS

We present statistics of the five commonsense reasoning (CSR) dataset we use in our experiments in Table 2.

## C  FULL EXPERIMENT RESULTS AND ANALYSIS

### C.1  MAIN RESULTS

We present results on GPT-2 in Table 3, T5 in Table 4, and Flan-T5 in Table 5. We do not use Flan-T5-XXL, which is too large (11B) to store on our hardware.

Table 2: Statistics of datasets

| Dataset Name | Type of CSR | Number of choices | Train | Validation | Test |
|---|---|---|---|---|---|
| COPA (Gordon et al., 2012) | Causal | 2 | N/A | 500 | 500 |
| CSQA (Talmor et al., 2019) | General | 5 | 9741 | 1221 | 1140 |
| OBQA (Mihaylov et al., 2018) | Scientific | 4 | 4957 | 500 | 500 |
| PIQA (Bisk et al., 2019) | Physical | 2 | 16000 | 2000 | 3000 |
| SIQA (Sap et al., 2019) | Social | 3 | 33410 | 1954 | N/A |

**Calibration is the best method on GPT-2 and T5.** Calibration works well on OBQA, outperforming the second-best baseline by 10.6% and 3.5% absolute for GPT-2 and T5, and similarly on COPA, CSQA, and SIQA. This is because calibration mitigates the surface form competition (Holtzman et al., 2021) by factoring out the probability of surface forms.

**Prompt engineering is the best method on Flan-T5.** On SIQA, prompt engineering beats other baselines by 3.9-10.7% absolute, and similarly on COPA, CSQA, and OBQA. This is because Flan-T5 has been instruction-tuned on many NLP tasks (Chung et al., 2022), and some of them are written with multiple choice prompts. Another cause is Flan-T5 has seen the training splits of all the five commonsense reasoning benchmarks during instruction tuning.

**Neither strategy works on PIQA.** On all models, ZS is the strongest baseline on PIQA, and beats the second-best baseline by 3.9-7.5% absolute. We attribute this fact to that solving PIQA requires different commonsense knowledge and reasoning than other benchmarks. PIQA focuses on physical knowledge, like gravity, momentum, and force. On the other hand, other benchmarks are more human-centric, focusing on general and social commonsense. We believe these models are not good at physical commonsense, so calibration and prompt engineering degrade performance.

**The joint effects of the two strategies, i.e., FULL, are mostly negative.** In most cases, the performance of FULL roughly equals the summed effect of prompt engineering and calibration, which is intuitive. We also find FULL only performs best on OBQA and CSQA with Flan-T5, where it is only marginally better than PE. For Flan-T5, this is likely because instruction tuning does not work well on small models, so FULL is partially functional. For GPT-2 and T5, this is likely because they are not instruction tuned, so the longer context introduced by PE and FULL degrades performance.

Table 3: Accuracy (%) on GPT-2

| Model | GPT-2-Base (125M) | | | | GPT-2-Medium (350M) | | | | GPT-2-Large (765M) | | | | GPT-2-XL (1.6B) | | | |
|---|---|---|---|---|---|---|---|---|---|---|---|---|---|---|---|---|
| | ZS | CA | PE | FULL | ZS | CA | PE | FULL | ZS | CA | PE | FULL | ZS | CA | PE | FULL |
| COPA | 61.0 | **62.8** | 53.0 | 54.4 | 67.0 | **70.0** | 49.4 | 54.2 | **69.8** | 69.4 | 51.4 | 57.4 | 69.0 | **71.6** | 51.4 | 53.0 |
| CSQA | 25.5 | **36.4** | 23.8 | 27.4 | 30.9 | **41.8** | 27.4 | 30.1 | 33.3 | **44.5** | 26.9 | 33.2 | 38.6 | **47.8** | 35.1 | 36.2 |
| OBQA | 15.8 | **33.4** | 25.6 | 28.0 | 18.0 | **38.6** | 26.8 | 27.4 | 21.6 | **41.4** | 25.2 | 29.4 | 22.4 | **43.2** | 25.8 | 29.4 |
| PIQA | **62.1** | 57.1 | 54.6 | 52.6 | **66.2** | 57.5 | 51.8 | 52.6 | **69.6** | 60.7 | 55.0 | 54.6 | **69.6** | 62.2 | 52.6 | 53.4 |
| SIQA | 35.8 | **38.0** | 34.3 | 37.1 | 36.9 | **40.0** | 36.0 | 38.0 | 36.6 | **40.3** | 34.0 | 35.6 | 39.0 | **41.0** | 35.2 | 35.9 |

Table 4: Accuracy (%) on T5

| Model | T5-Small (80M) | | | | T5-Base (250M) | | | | T5-Large (780M) | | | |
|---|---|---|---|---|---|---|---|---|---|---|---|---|
| | ZS | CA | PE | FULL | ZS | CA | PE | FULL | ZS | CA | PE | FULL |
| COPA | **55.2** | 51.2 | 51.2 | 52.2 | **59.6** | 59.4 | 51.0 | 51.8 | **65.2** | 56.6 | 53.2 | 53.8 |
| CSQA | 16.6 | **22.8** | 21.1 | 21.0 | 26.1 | **30.0** | 20.6 | 22.5 | **39.2** | 35.4 | 33.1 | 35.7 |
| OBQA | 14.2 | **28.8** | 23.8 | 25.8 | 15.8 | **30.8** | 27.8 | 27.2 | 19.0 | **30.4** | 24.8 | 26.4 |
| PIQA | 56.6 | 50.5 | 51.2 | 50.8 | **61.0** | 57.7 | 51.7 | 53.0 | **66.6** | 64.4 | 52.8 | 51.7 |
| SIQA | **36.2** | 36.1 | 35.0 | 34.4 | 36.2 | **37.6** | 37.0 | 33.5 | **38.7** | 38.1 | 37.0 | 34.1 |

Table 5: Accuracy (%) on Flan-T5

| Model | Flan-T5-Small (80M) | | | | Flan-T5-Base (250M) | | | | Flan-T5-Large (780M) | | | | Flan-T5-XL (3B) | | | |
|---|---|---|---|---|---|---|---|---|---|---|---|---|---|---|---|---|
| | ZS | CA | PE | FULL | ZS | CA | PE | FULL | ZS | CA | PE | FULL | ZS | CA | PE | FULL |
| COPA | **59.8** | 56.6 | 52.0 | 49.6 | 67.0 | **68.2** | 60.6 | 61.4 | 72.8 | 71.6 | **87.6** | 84.0 | 80.8 | 78.4 | **88.8** | 85.6 |
| CSQA | 29.2 | **37.7** | 30.8 | 28.3 | 40.9 | 48.5 | **52.5** | 51.8 | 51.6 | 51.5 | 62.2 | **67.6** | 61.8 | 64.7 | 70.6 | **72.7** |
| OBQA | 14.0 | **32.6** | 24.8 | 29.6 | 20.0 | **34.0** | 28.6 | 34.0 | 24.2 | 39.4 | **53.4** | 52.8 | 30.0 | 49.6 | **61.0** | 55.4 |
| PIQA | **62.5** | 57.6 | 54.2 | 51.1 | **65.9** | 59.7 | 58.1 | 54.0 | 71.4 | 65.5 | **72.7** | 60.6 | **75.8** | 68.3 | 68.9 | 60.4 |
| SIQA | 41.7 | **42.5** | 42.3 | 42.3 | 46.4 | 47.4 | **54.7** | 53.7 | 51.4 | 48.1 | **68.6** | 66.7 | 56.1 | 56.3 | **71.6** | 58.9 |

## C.2 ABLATION STUDY: THE EFFECT OF SYMBOL BINDING

We apply symbol binding (SB) to PE and FULL, and compare their accuracy (%) on SIQA. Results are shown in Table 6. We find symbol binding universally decreases performance on all models and on both methods, which is likely because symbol binding is an emergent ability that only benefits large models (Robinson & Wingate, 2023). Therefore, we do not use symbol binding in our experiments.

Table 6: Effect of symbol binding (%) on SIQA

| Method | GPT-2 | T5 | Flan-T5 |
|---|---|---|---|
| PE | 34.9 | 36.3 | 59.3 |
| PE + SB | 33.0 (-1.9) | 32.6 (-3.7) | 57.8 (-1.5) |
| FULL | 36.7 | 34.0 | 55.4 |
| FULL + SB | 33.6 (-3.1) | 33.1 (-0.9) | 52.0 (-3.4) |

