# OpenReview forum: "Prompt Engineering and Calibration for Zero-Shot Commonsense Reasoning"
_ICLR.cc/2023/TinyPapers — Submitted to Tiny Papers @ ICLR 2023_

### Official Review · Reviewer_PTuR · 2023-03-31

**Confidence:** 4

**Summary Of Contributions:**

This paper investigates whether prompt engineering and calibration help smaller language models in multiple choice commonsense reasoning. The key findings are that each strategy favors some language models and their joint effects are mostly negative.

**Rating:**

Great Start (GS): a submission which meets some of the reviewing criteria but has room for improvement

**Strengths And Weaknesses:**

Strengths:

1. This work studies the interesting topic of how to make small models benefit from advanced techniques like prompt engineering and calibration, as they help LLMs to perform reasoning. The topic would be of interest to the community.

2. Comprehensive experiments are conducted on five popular multiple-choice commonsense benchmarks. The results are convincing.

Weaknesses:

1. Some descriptions need more clarity. For example, In Figure 1 caption, "Unlike previous works, we do not bind options to symbols, like (A)." Why did the authors make this choice instead of following previous works? How about the influence on the final performance?

2. According to the results presented in Tables 1, Table 4, and Table 5, the two approaches have different effects on the performance. The authors thus concluded that "each strategy favors some language model"; however, the conclusion is too vague to give insights to readers. A more in-depth analysis of why the models can benefit from different strategies would make this work more interesting. I give some examples below to show my thoughts on interpreting the results. They are not blames but are suggested analyses in the future study.
For example, (i) we see that CA (which is ZS with calibration) generally works well with GPT-2. Does this indicate that CA is more useful for auto-regressive models?; (ii) PE becomes more effective with larger Flan-T5 models. This aligns with the existing findings of emergent ability as larger models have a better ability for in-context learning; (iii) ZS performs the best in vanilla T5 models compared with the other prompting strategies, which tends to indicate that vanilla T5 models can hardly learn from the prompts.



**Suggested Changes:**

Please see the weakness part above. The paper can be improved with better clarity and a more in-depth analysis of why different techniques favor different models.

---

> ### Author Response · Authors · 2023-05-19
> **Reponse to review PTuR**
>
> Thank you for your comments. Please find our response below.
>
> >Why did the authors make this choice instead of following previous works? How about the influence on the final performance?
>
> We apply symbol binding (SB) to PE and FULL, and compare their accuracy (\%) on SIQA. Results are shown in the following table. We find symbol binding universally decreases performance on all models and on both methods, which is likely because symbol binding is an [emergent ability](https://openreview.net/forum?id=yKbprarjc5B) that only benefits large models. Therefore, we do not use symbol binding in our experiments.
>
> | Method     | GPT-2 | T5   | Flan-T5 |
> |------------|-------|------|---------|
> | PE         | 34.9  | 36.3 | 59.3    |
> | PE + SB    | 33.0 (-1.9) | 32.6 (-3.7) | 57.8 (-1.5) |
> | FULL       | 36.7  | 34.0 | 55.4    |
> | FULL + SB  | 33.6 (-3.1) | 33.1 (-0.9) | 52.0 (-3.4) |
>
> > the conclusion is too vague to give insights to readers.
>
> We have revised our conclusion to include the following takeaways:
> 1. Calibration is the best method on GPT-2 and T5.
> 2. Prompt engineering is the best method on Flan-T5 models.
> 3. Their joint effects are generally negative.
>
> > A more in-depth analysis of why the models can benefit from different strategies
>
> Calibration is the best method on GPT-2 and T5. It works well on OBQA, outperforming the second-best baseline by 10.6\% and 3.5\% absolute for GPT-2 and T5, and similarly on COPA, CSQA, and SIQA. This is because calibration mitigates the [surface form competition](https://aclanthology.org/2021.emnlp-main.564/) by factoring out the probability of surface forms.
>
> Prompt engineering is the best method on Flan-T5. On SIQA, prompt engineering beats other baselines by 3.9-10.7\% absolute, and similarly on COPA, CSQA, and OBQA. This is because Flan-T5 has been [instruction-tuned](https://arxiv.org/abs/2210.11416) on many NLP tasks, and some of them are written with multiple choice prompts. Another cause is Flan-T5 has seen the training splits of all the five commonsense reasoning benchmarks during instruction tuning.
>
> We also find the joint effects of the two strategies, i.e., FULL, are mostly negative. In most cases, the performance of FULL roughly equals the summed effect of prompt engineering and calibration, which is intuitive. We also find FULL only performs best on OBQA and CSQA with Flan-T5, where it is only marginally better than PE.
> For Flan-T5, this is likely because instruction tuning does work well on small models, so FULL is partially functional. For GPT-2 and T5, this is likely because they are not instruction tuned, so the longer context introduced by PE and FULL degrades performance.
>
> > better clarity and a more in-depth analysis of why different techniques favor different models.
>
> We have added analysis in Appendix C.1, and discussion on symbol binding in Appendix C.2.

---

### Official Review · Reviewer_wFpt · 2023-04-02

**Confidence:** 4

**Summary Of Contributions:**

The authors investigate the applicability of prompt engineering and calibration techniques on smaller language models (up to 3B parameters) for zero-shot multiple choice commonsense reasoning tasks. Based on experiments from five commonsense reasoning benchmarks, the results reveal that each strategy benefits certain models, but their combined effects are mostly negative.

**Rating:**

Clear, Correct, and Reproducible (CCR): a submission which meets the reviewing criteria

**Strengths And Weaknesses:**

Strengths:

* The paper clearly states the motivation, the specific experiments used for testing, and provides comprehensive experiment results that are reproducible.
* The experiments from multiple models clearly support the conclusions made on how joint effects are mostly negative and the best approach is model specific.
* The paper is well written with a clear flow of thoughts from the beginning to the end.

Weaknesses: There are no apparent weaknesses, I think analysis of small models are still relevant despite what a lot of people believe.



**Suggested Changes:**

It's probably easier for the reader if the url to the code is included within in the first two pages of the paper instead of the appendix.

---

> ### Author Response · Authors · 2023-05-19
> **Reponse to review wFpt**
>
> Thank you for your comments. Please find our response below.
>
> >It's probably easier for the reader if the url to the code is included within in the first two pages of the paper instead of the appendix.
>
> We have open sourced [our code](https://github.com/KasMasVan/Prompt-engineering-and-calibration). The url to the code is now on the first page as a footnote.

---

### Official Review · Reviewer_m239 · 2023-04-04

**Confidence:** 3

**Summary Of Contributions:**

This paper tries to evaluate the impact of prompt engineering and calibration techniques on smaller LLMs at reasoning tasks. Through their experimentations, they demonstrate that each strategy favours certain model and their joint effect is usually negative.

**Rating:**

Great Start (GS): a submission which meets some of the reviewing criteria but has room for improvement

**Strengths And Weaknesses:**

Strengths:
* The authors in this paper try to understand the impact of calibration and prompt engineering on small LLMs and use a set of commonsense reasoning benchmarks for that evaluation.
*  The authors demonstrate the idea of calibration and prompt engineering really well and concisely in their figure, which can easily be understood by the reader.

Weakness:
* The paper tries to evaluate the impact of calibration and prompt engineering, but no conclusive takeaways are provided in the paper. These strategies do not seem to follow a certain pattern, in that case, the authors should have tried to understand why a certain approach works on a certain dataset.

**Suggested Changes:**

* The authors can try to dive deep into understanding the reasons for the performance of different models. An in-depth analysis will bring more nuance to this paper.

---

> ### Author Response · Authors · 2023-05-19
> **Reponse to review m239**
>
> Thank you for your comments. Please find our response below.
> >no conclusive takeaways
>
> We summarize four takeaways from our experiments (and have revised our paper accordingly):
> 1. Calibration is the best method on GPT-2 and T5.
> 2. Prompt engineering is the best method on Flan-T5 models.
> 3. Neither strategy works on PIQA.
> 4. Their joint effects are generally negative.
>
> > why a certain approach works on a certain dataset.
>
> Calibration is the best method on GPT-2 and T5. It works well on OBQA, outperforming the second-best baseline by 10.6\% and 3.5\% absolute for GPT-2 and T5, and similarly on COPA, CSQA, and SIQA. This is because calibration mitigates the [surface form competition](https://aclanthology.org/2021.emnlp-main.564/) by factoring out the probability of surface forms.
>
> Prompt engineering is the best method on Flan-T5. On SIQA, prompt engineering beats other baselines by 3.9-10.7\% absolute, and similarly on COPA, CSQA, and OBQA. This is because Flan-T5 has been [instruction-tuned](https://arxiv.org/abs/2210.11416) on many NLP tasks, and some of them are written with multiple choice prompts. Another cause is Flan-T5 has seen the training splits of all the five commonsense reasoning benchmarks during instruction tuning.
>
> We also find neither strategy works on PIQA. On all models, ZS is the strongest baseline on PIQA, and beats the second best by 3.9-7.5\% absolute. We attribute this fact to that solving PIQA requires different commonsense knowledge and reasoning than other benchmarks. PIQA focuses on physical knowledge, like gravity, momentum, and force. On the other hand, other benchmarks are more human-centric, focusing on general and social commonsense. We believe these models are not good at physical commonsense, so calibration and prompt engineering degrade performance.
>
> > dive deep into understanding the reasons for the performance of different models
>
> We have added analysis in Appendix C.1.

---

### Author Response · Authors · 2023-05-30
**Opt-in for Archival**

We wish to opt-in for archival.

---

### Meta-Review · Area_Chair_6Xai · 2023-04-05

**Recommendation:** Invite to archive
**Confidence:** 4

**Metareview:**

Overall, this work makes valuable attempts to evaluate the impact of prompt engineering and calibration techniques on smaller LLMs at reasoning tasks. The topic would be of interest to the community.

Pros: The paper is well-written and easy to read. The idea of calibration and prompt engineering is clearly demonstrated. The experimental results are convincing.

Cons: Two reviewers mentioned a lack of conclusive takeaways provided in the paper. Reviewer PTuR provided detailed suggestions for the authors to interpret the results.



**Summary:**

This work investigates the effects of prompt engineering and calibration on small language models on multiple choice commonsense reasoning. The reviewers noted the strengths of comprehensive experiments and good writing quality. However, it is hard to interpret new findings from the results.

**Comments And Feedback To The Authors:**

This work would be more interesting if the authors can provide a more in-depth analysis following the reviewers' suggestions.

**Reason For Not Giving A Higher Recommendation:**

There is a need for more thorough consideration of the experiment organization. More concrete conclusions could be reached to help readers understand why a certain approach works on a certain dataset.

**Reason For Not Giving A Lower Recommendation:**

N/A

---

### Decision · Program_Chairs · 2023-04-10

Invite to archive